# What We Know about the Long-Term Risks of Hysterectomy for Benign Indication—A Systematic Review

**DOI:** 10.3390/jcm10225335

**Published:** 2021-11-16

**Authors:** Obianuju Sandra Madueke-Laveaux, Amro Elsharoud, Ayman Al-Hendy

**Affiliations:** 1Department of Obstetrics and Gynecology, University of Chicago, Chicago, IL 60637, USA; slaveaux@BSD.uchicago.edu; 2Department of Family Medicine, Texas Tech University Health and Science Center, Lubbock, TX 79430, USA; amro.elsharoud@ttuhsc.edu

**Keywords:** uterine fibroids, long-term consequences of hysterectomy, uterine fibroid management, patient engagement, complications, adverse events

## Abstract

Hysterectomy is the most common treatment option in women with uterine fibroids, providing definitive relief from the associated burdensome symptoms. As with all surgical interventions, hysterectomy is associated with risk of complications, short-term morbidities, and mortality, all of which have been described previously. However, information on the potential long-term risks of hysterectomy is only recently becoming available. A systematic literature review was performed to identify studies published between 2005 and December 2020 evaluating the long-term impact of hysterectomy on patient outcomes. A total of 29 relevant studies were identified. A review of the articles showed that hysterectomy may increase the risk of cardiovascular events, certain cancers, the need for further surgery, early ovarian failure and menopause, depression, and other outcomes. It is important to acknowledge that the available studies examine possible associations and hypotheses rather than causality, and there is a need to establish higher quality studies to truly evaluate the long-term consequences of hysterectomy. However, it is of value to consider these findings when discussing the benefits and risks of all treatment options with patients with uterine fibroids to allow for preference-based choices to be made in a shared decision-making process. This is key to ensuring that patients receive the treatment that best meets their individual needs.

## 1. Introduction

Uterine fibroids (UF), often referred to as leiomyomas or myomas, are common benign tumors of the uterus [1]. UF are dependent on estrogen and progesterone for their growth and tend to arise during women’s reproductive years and regress to varying degrees after menopause [2,3,4]. The incidence of UF increases as women age, with an estimated cumulative incidence of >80% for African American women and nearly 70% for Caucasian women reported by 50 years of age [5]. Other risk factors for developing UF include the following demographics: Black race, nulliparity, obesity, family history of UF, and hypertension [1,2,4].

Although the majority of UF are asymptomatic, in approximately 25% of women they cause symptoms that may require treatment [1,6]. Potential symptoms of UF include heavy menstrual bleeding (HMB), pain, and “bulk”-related symptoms caused by large myoma(s), such as pressure in the abdomen and pelvis, urinary symptoms (urinary urgency, frequency, and incontinence), and bowel symptoms (constipation and tenesmus) [7,8]. Persistent HMB can induce iron-deficiency anemia [1] and associated fatigue [9].

In addition to the adverse physical effects of symptomatic UF, women’s psychological and emotional health is also impacted, leading to depression, social isolation, feelings of helplessness, and anxiety [10]. Symptoms of UF can severely impair multiple aspects of women’s health-related quality of life, including productivity, sexuality, physical and emotional wellbeing, and relationships [8]. Thus, the negative effects of UF have broad and far-reaching implications on women’s social, work/school, and family lives [2,11,12].

There is much available information in the literature about treatment options for symptomatic UF, which can be broadly categorized into one of three types: (1) surgery, (2) non-surgical/minimally invasive procedures (e.g., uterine artery embolization, magnetic resonance-guided focused ultrasound, and radiofrequency ablation), or (3) medical treatment. Hysterectomy (the surgical removal of the uterus) is one of the main treatment options, especially in women with UF for whom family planning is no longer a consideration [13]. It is the most commonly used surgical option in women with UF worldwide [14] and is estimated to account for 66.8–76.5% of all surgical and minimally invasive procedures for UF according to a 2013 Healthcare Cost and Utilization Project Statistical Brief, which examined data from 13 US states [15]. Hysterectomy is the only definitive treatment option for UF, providing complete and permanent relief from the extreme burden of HMB, pain, and other UF-associated symptoms [13,16]. In some patients with UF, hysterectomy is difficult to exclude—for example, when there is suspicion of malignancy; when medical management fails; when fibroids are large (increasing uterus size to ≥20-week-size uterus); or when fibroids lead to organ damage/insult (e.g., ureteral compression; vessel compression leading to lower extremity clots). Substantial improvements in short-term health-related quality of life, assessed using the disease-specific Uterine Fibroid Symptom Quality of Life and the generic EuroQoL 5-Dimension Health Questionnaire 6–12 weeks after surgery, have also been reported for women with UF who underwent hysterectomy [17]. 

Surgical complications, mortality risks, and various short-term morbidities, including increased risk of intra-abdominal adhesions, postoperative infections, pelvic organ dysfunction, and thromboembolic events associated with hysterectomy, are well described [18,19,20,21]. Rates of major morbidities and mortality for the abdominal and laparoscopic hysterectomy routes are shown in (Table 1). The abdominal route of hysterectomy is associated with increased risk of morbidity compared with the laparoscopic or vaginal route [22]. Concerns related to the power morcellation of uterine specimens and the associated risk of spreading cancerous tissue beyond the uterus led to the Food and Drugs Administration warning against the use of power morcellation for women undergoing minimally invasive hysterectomy for UF in 2014 [23]. Following the warning, the rates of laparoscopic hysterectomy (and myomectomy) were reported to decrease, whereas use of the abdominal route for both procedures increased [24,25], leading to increases in major and minor 30-day complications and 30-day readmissions following hysterectomy [24,26]. However, this was disputed in a retrospective analysis in which rates of laparoscopic supracervical hysterectomy were reported to decrease, but the rates of other laparoscopic hysterectomies increased, and patient outcomes were not impacted overall [27]. Bilateral oophorectomy surgery was previously reported to be common practice during hysterectomy to avoid risk of future surgery or ovarian cancer. However, estrogen deficiency from pre- and post-menopausal oophorectomy may be associated with increased risk of coronary heart disease, stroke, hip fracture, and cognitive disease, and the concerns that women will require a subsequent oophorectomy following a hysterectomy are unfounded [28,29]. Therefore, where possible, whether bilateral or unilateral oophorectomy occurred was considered in this systematic literature review. The long-term risks associated with hysterectomy are less well-known. Considering that hysterectomy is the most widely used intervention for UF, the purpose of this publication is to examine the available data on the potential long-term complications of its use [14].

## 2. Methods

A systematic literature review was performed to identify studies evaluating the long-term impact of hysterectomy on patient outcomes, including cardiovascular events, cancer, the need for further surgery, early ovarian failure and menopause, depression, and other outcomes. The process was fully compliant with PRISMA guidelines from searches through to data extraction [30]. Searches were performed using three electronic databases: MEDLINE, Embase, and the Cochrane library. The search strings (Appendix A) used to identify relevant evidence included free text and Medical Subject Headings (MeSH) terms. Studies were limited to those published between January 2005 and December 2020 with at least 100 subjects (to ensure robustness of outcomes and to minimize selection bias). Relevant studies from all countries were eligible. A comparator group comprising women with either no hysterectomy, an external control group or healthy controls was required. It was not a requirement for UF to be the sole reason for hysterectomy; indications for hysterectomy included the presence of UF or other benign conditions. To identify potentially relevant studies, abstracts and titles were assessed by a single reviewer in a single-blind process with reference to the criteria listed in Table 2. Full-text articles were obtained where titles and/or abstracts were deemed to be relevant or where eligibility was unclear. The reference sections of relevant reviews included during citation-screening were searched for potentially relevant studies. A narrative synthesis of the data was carried out. There was no formal method of combining individual study data using statistical methods. The outcomes in each study were summarized using descriptive methods but were not compared across studies. 

The electronic database search identified 2143 articles in total, 359 of which were duplicates and removed, resulting in 1784 articles for screening. After applying the inclusion and exclusion criteria, 1634 articles were excluded. A total of 150 articles were obtained for a full reference review. Of these, 121 were excluded on full review, leaving 30 articles (Appendix A and Table 3).

## 3. Results

The studies identified from the literature search are summarized in Table 3, and the impact of the various long-term outcomes in women with or without hysterectomy from each study is subsequently discussed in the sections below. 

**Table 3 jcm-10-05335-t003:** Summary of studies investigating long-term outcomes in women with or without hysterectomy.

Author	Study Type	Number of Women	Duration of Observation after Hysterectomy	Outcomes Assessed	Outcomes (Risk Increased [↑], Deceased [↓] or Not Significantly [NS] Changed with Hysterectomy vs. No Hysterectomy)
**Mortality, cardiovascular disease, hypertension, and stroke**					
Wilson et al., 2019 [31]	Population-based cohort study	H (oo+): 2472 H (oo−): 851 w/o H: 10,218	21.5 years (median)	All-cause mortality risk	HR; 95% CI H (oo+): 0.86; 0.72 to 1.02 NS H (oo−): 1.02; 0.78 to 1.34 NS Women who did not use menopausal hormone therapy: H (oo−) before age 50 years: 1.81; 1.01 to 3.25 **↑**
Iversen et al., 2005 (see cancer section) [32]	Nested cohort study	H: 3705 w/o H: 3705	250.3 months (mean)	Long-term risk of death from all causes, CVD, and cancer	aHR; 95% CI ≤43.7 years All-cause mortality: 0.82; 0.65 to 1.03 NS CVD: 0.85; 0.54 to 1.33 NS >43.7 years All-cause mortality: 0.94; 0.75 to 1.18 NS CVD: 0.80; 0.52 to 1.23 NS
Howard et al., 2005 [33]	Observational study	H (oo+/oo−): 36,865 (approx. half underwent oo−) w/o H: 53,409 (calculated)	5.1 years (mean)	CVD risk	HR 1.26; 95% CI 1.16 to 1.36; *p* < 0.001 **↑** After adjustment for demographic variables and CVD risk factors NS
Laughlin-Tommaso et al., 2018 [34]	Population-based cohort study	H: 2094 w/o H: 2094	H: 22.5 years (median) w/o H: 21.3 years (median) Cohorts combined: 21.9 years (median)	Long-term risk of de novo cardiovascular and metabolic conditions	Hyperlipidemia: aHR 1.14; 95% CI 1.05 to 1.25 **↑** Hypertension: aHR 1.13; 95% CI 1.03 to 1.25 **↑** Obesity: aHR 1.18; 95% CI 1.04 to 1.35 **↑** Cardiac arrhythmias: aHR 1.17; 95% CI 1.05 to 1.32 **↑** CAD: aHR 1.33; 95% CI 1.12 to 1.58 **↑**
Ingelsson et al., 2011 [35]	Population-based cohort study	H: 184,441 w/o H: 640,043	10.4 years (median)	First hospitalization or death of incident CVD (coronary heart disease, stroke, heart failure)	Age ≤ 49 years (aHR; 95% CI) H (oo+): 1.18; 1.13 to 1.23 **↑** H (oo− before study entry): 2.22; 1.01 to 4.83 **↑** H (oo− after study entry): 1.25; 1.06 to 1.48 **↑**
Choi & Lee 2018 [36]	Population-based cohort study	H: 11,280 w/o H: 45,120	H: 74.3 months (mean) w/o H: 72.3 (mean)	Stroke risk	Hemorrhagic stroke: aHR 0.91; *p* = 0.592 NS Ischemic stroke: aHR 0.85; *p* = 0.188 NS
Yeh et al., 2013 [37]	Population-based cohort study	H: 7605 w/o H: 30,420	7.24 years (median)	Stroke risk	Stroke (hysterectomy before age 45 years): HR 2.29; 95% CI 1.52 to 3.44 **↑**
Ding et al., 2018 [38]	Population-based cohort study	H: 7331 w/o H: 29,324	H: 7.0 years (mean) w/o H: 7.1 years (mean)	CAD risk	aHR 1.31; 95% CI 1.18 to 1.45; *p* < 0.001 **↑**
Ding et al., 2018 [39]	Population-based cohort study	H: 6674 w/o H: 26,696	H: 6.1 years (median)w/o H: 6.5 years (median)	Hypertension risk	aHR 1.35; 95% CI 1.27 to 1.44 **↑**
Li et al., 2018 [40]	Population-based cohort study	H: 5887 w/o H: 28,024	H: 5.95 (mean) w/o H: 5.96 (mean)	Hyperlipidemia risk	aHR 1.27; 95% CI 1.19 to 1.35 **↑**
**Cancer**					
Iversen et al., 2005 (see CVD section) [32]	Nested cohort study	H: 3705 w/o H: 3705	250.3 months (mean)	Long-term risk of death from all causes, CVD, and cancer	aHR; 95% CI ≤43.7 years Cancer: 0.81; 0.55 to 1.19 NS >43.7 years Cancer: 1.02; 0.69 to 1.49 NS
Altman et al., 2010 [41]	Population-based cohort study	H: 184,945 w/o H: 657,288	H: 2,061,556 person-years w/o H: 7,631,824 person-years	RCC, urinary tract cancer, bladder cancer risk	RCC: HR 1.50; 95% CI 1.33 to 1.69 **↑** Urinary tract: HR 1.24; 95% CI 0.88 to 1.75 **↑** Bladder: HR 1.21; 95% CI 1.07 to 1.37 **↑**
Guenego et al., 2019 [42]	Prospective cohort study	H: 9143 w/o H: 80,197	Follow-up started at the date of return of the 1990 questionnaire. Participants contributed person-years of follow-up until the date of diagnosis of any cancer (except basal cell carcinoma and in situ colorectal cancer), the date of the last completed questionnaire, or December 2011	Thyroid cancer risk	History of hysterectomyaHR 2.05; 95% CI 1.65 to 2.55 **↑** History of fibroids aHR 1.91; 95% CI 1.50 to 2.44 **↑**
Luo et al., 2016 [43]	Prospective cohort study	H: 24,575 H w/o: 43,139	Participants were followed up from enrollment to first thyroid cancer diagnosis, date of death, loss to follow-up, or end of clinical trial or observational study follow-up (30 September 2015), whichever occurred first	Thyroid cancer risk	HR 1.46; 95% CI 1.16 to 1.85 **↑**
Falconer et al., 2017 [44]	Population-based cohort study	H: 52 w/o H: 2882	H: 500,698 person-years w/o H: 54,988,227 person-years	Thyroid cancer risk	Papillary thyroid cancer: HR 1.70; 95% CI 1.04 to 2.79 **↑** Follicular or “other” subtypes: NS
**Incontinence, pelvic prolapse, pelvic organ fistula, lower urinary tract infection**					
Forsgren et al., 2009 [45]	Population-based cohort study	H: 182,641 w/o H: 525,826	H: 1,970,076 person-years w/o H: 6,114,023 person-years	Pelvic organ fistula disease risk	HR 3.8; 95% CI 3.3 to 4.3 **↑**
Altman et al., 2008 [46]	Population-based longitudinal cohort study	H: 162,488 w/o H: 470,519	H: 11.9 years (mean) w/o H: 12.1 years (mean)	Risk for pelvic organ prolapse surgery	HR 1.7; 95% CI 1.6 to 1.7 **↑**
Altman et al., 2007 [47]	Population-based cohort study	H: 165,260 w/o H: 479,506	H: 11.9 years (mean) w/o H: 12.1 years (mean)	Short-term and long-term risk for stress-urinary-incontinence surgery	HR 2.4; 95% CI 2.3 to 2.5 **↑**
Li et al., 2019 [48]	Population-based cohort study	H: 8514 w/o H: 34,056	Follow-up from the index date to the occurrence of lower urinary tract symptoms, death, withdrawal from the program or 31 December 2013 7.7 years (median)	Effect on de novo lower urinary tract symptoms	aHR 1.57; 95% CI 1.46 to 1.70 **↑**
**Premature menopause and ovarian failure, health status, osteoporosis, frailty, vasomotor symptoms**					
Moorman et al., 2011 [49]	Prospective cohort study	H: 406 w/o H: 465	Follow-up from index date to November 2009	Risk for earlier ovarian failure	HR 1.92; 95% CI 1.29 to 2.86 **↑**
Farquhar et al., 2005 [50]	Prospective cohort study	H: 257 w/o H: 259	5 years’ follow-up	Influence on ovarian function	Reached menopause during the 5-year follow-up period: 20.6% vs. 7.3% **↑**
Verschoor & Tamim 2019 [51]	Cross-sectional cohort study	H: 2182 w/o H: 7379	Cross-sectional study	Frailty	Adjusted OR 1.59; 95% CI 1.25 to 2.02; *p* < 0.001 **↑**
Farquhar et al., 2008 [52]	Prospective cohort study	H: 257 w/o H:257	Follow-up at 5 years	Gynecological, abdominal, urinary symptoms and sexual functioning, depression, and self-rated health 5 years after hysterectomy	Hot flushes (41% vs. 19%; *p* = 0.05) **↑** Vaginal dryness (48% vs. 20%; *p* < 0.004) **↑**
Choi et al., 2019 [53]	Prospective cohort study	H: 9082 w/o H: 36,328	H: 63.0 months (mean) w/o H: 66.9 months (mean)	Osteoporosis occurrence	HR 1.45; 95% CI 1.37 to 1.53; *p* < 0.001 **↑**
Wilson et al., 2016 [54]	Population-based cohort study	H: 1129 w/o H: 4977	17 years	Symptom patterns for hot flushes and night sweats	Constant vs. minimal hot flushes: OR 1.97; 95% CI 1.64 to 2.35 **↑** Constant vs. minimal night sweats OR 2.09; 95% CI 1.70 to 2.55 **↑**
**Depression, dementia, cognitive function**					
Choi et al., 2020 [55]	Population-based cohort study	H: 9971 w/o H: 39,884	H: 72.7 months (mean) w/o H: 73.1 months (mean)	Influence on depression	HR 1.15; 95% CI 1.03 to 1.29; *p* < 0.05 **↑**
Wilson et al., 2018 [56]	Population-based cohort study	H (oo+): 884 H (oo−): 450 w/o H: 4002	12 years	Incidence of depressive symptoms	H (oo+): RR 1.20; 95% CI 1.06 to 1.36 **↑** H (oo−): RR 1.44; 95% CI 1.22 to 1.68 **↑**
Laughlin-Tommaso et al., 2020 [57]	Population-based cohort study	H: 2094 w/o H: 2094	H: 22.5 years (median) w/o H: 21.3 years (median) Both cohorts combined:21.9 years (median)	Long-term associations with a broad range of aging-related mental health conditions	De novo depression: aHR 1.26; 95% CI 1.12 to 1.41 **↑** De novo anxiety: aHR 1.22; 95% CI 1.08 to 1.38 **↑**
Phung et al., 2010 [58]	Population-based cohort study	H: 3534 w/o H: 91,705	Study population was followed from 1 January 1977, or age 40 years, whichever came later, until the date of dementia onset, date of death, date of emigration or 31 December 2006, whichever came first	Early-onset dementia risk	Early-onset dementia: RR 1.38; 95% CI 1.07 to 1.78 **↑**
Shen et al., 2019 [59]	Population-based cohort study	H (oo−): 4337 w/o H: 17,348	H: 7.93 years (mean) w/o H: 7.93 years (mean)	Bipolar disorder risk	Adjusted IRR 2.19; 95% CI 1.94 to 2.49 **↑**

aHR, adjusted hazard ratio; CAD, coronary artery disease; CI, confidence interval; CVD, cardiovascular disease; H, hysterectomy; HR, hazard ratio; IRR, incidence rate ratio; oo+, ovarian conservation; oo−, bilateral oophorectomy; OR, odds ratio; RCC, renal cell carcinoma; RR, relative risk; w/o H, without hysterectomy.

### 3.1. Impact of Hysterectomy on Mortality, Cardiovascular Disease, Hypertension, and Stroke

In total, nine studies were identified that investigated the association between hysterectomy and risk of cardiovascular disease (CVD), metabolic conditions, or stroke [24,32,33,34,35,36,37,38,39]. Six studies were also identified in which the association between hysterectomy and mortality was investigated. None of these six studies found an association between overall mortality risk and hysterectomy [31,32,33,34,37,44]. However, Wilson et al. reported that, among women who were non-users of menopausal hormonal replacement therapy, there was increased risk of death if they had undergone hysterectomy and bilateral oophorectomy before age 50 years compared with those who had not (hazard ratio (HR) 1.81; 95% confidence interval (CI) 1.01 to 3.25) [31].

Laughlin-Tommaso et al. conducted a long-term (1980–2002) investigation of associated risk between hysterectomy and CVD or metabolic conditions using data from the Rochester Epidemiology Project records-linkage system [34]. Women who had undergone hysterectomy with bilateral ovarian conservation (*n* = 2094) were age-matched with referent women (control) who had not had a hysterectomy or oophorectomy (*n* = 2094). The analysis was adjusted for 20 pre-existing chronic conditions present at the time of hysterectomy and other potential confounders including years of education (≤12, 13–16, >16 m unknown), race (white vs. non-white), and age and calendar year at baseline (continuous). Hysterectomy was performed for benign indications and UF accounted for 39.5% of these cases. Women who underwent hysterectomy experienced an increased risk of de novo hyperlipidemia (adjusted HR (aHR) 1.14; 95% CI 1.05 to 1.25), hypertension (aHR 1.13; 95% CI 1.03 to 1.25), obesity (aHR 1.18; 95% CI 1.04 to 1.35), cardiac arrhythmias (aHR 1.17; 95% CI 1.05 to 1.32), and coronary artery disease (CAD) (aHR 1.33; 95% CI 1.12 to 1.58) compared with those who did not undergo hysterectomy (Table 3). As shown in Figure 1, the risk of congestive heart failure and CAD increased by 4.6-fold and 2.5-fold in, respectively, women who underwent hysterectomy at age ≤35 years vs. age-matched women without hysterectomy [34]. 

In women who underwent hysterectomy with ovarian conservation specifically for the management of UF, the risks of de novo hyperlipidemia (*p* = 0.004), cardiac arrhythmias (*p* = 0.007), and hypertension (*p* = 0.07) also increased compared with referent women [34]. It is also worth noting that although no association between mortality and hysterectomy was observed in this study, the authors stated they will continue to follow the cohorts for future analyses of mortality. They considered that the relatively young age of the cohorts at the time of analysis precluded any effective assessment of mortality.

In contrast, an earlier investigation by Howard et al. using data from the Women’s Health Initiative Observational Study found that although hysterectomy was a significant predictor of CVD, the effect was reduced and nonsignificant when adjusted for demographic variables (such as age, ethnicity, family history, income, and education) and CVD risk factors [33]. However, the duration of postoperative follow-up was far shorter in this study (mean: 5.1 years) [33] than in the Rochester Epidemiology Project (median: 21.9 years) [34], making comparisons difficult. In addition, in a study of the Swedish Inpatient Registry (median follow-up 10.4 years), the risk of first hospitalization for, or death from, incident CVD (coronary heart disease, stroke, or heart failure) was shown to be significantly increased in women (age 18–49 years) with hysterectomy and without oophorectomy versus women (age 18–49 years) without hysterectomy: HR 1.18; 95% CI 1.13 to 1.23 [35].

Three population-based retrospective cohort studies were conducted using information from the Taiwan National Health Insurance Research Database (2000–2013) to investigate the association between hysterectomy and risk of de novo hypertension [39], CAD [38], and hyperlipidemia [24]. In all three studies, the risk of these adverse outcomes was greater among women who had undergone hysterectomy. The occurrence of incident hypertension was higher in women (age 30–49 years) with hysterectomy (*n* = 1284) vs. those without prior hysterectomy (*n* = 5166): aHR 1.35; 95% CI 1.27 to 1.44 [39]. An increased incidence of hypertension with age was also observed, with a higher risk observed for hysterectomized women vs. non-hysterectomized women aged 40–49 years than in those aged 30–39 years: aHR 1.37; 95% CI 1.06 to 1.83 and aHR 1.22; 95% CI 1.02 to 1.46, respectively. Similarly, a significant association was observed between hysterectomy and CAD with an incidence of 9.82 vs. 7.17/1000 person-years observed with hysterectomy (*n* = 7331) vs. without hysterectomy (*n* = 29,324) (aHR 1.31; 95% CI 1.18 to 1.45; *p* < 0.001) [38]. Finally, Li et al. observed an increased incidence (1.3 times greater) of de novo hyperlipidemia in women with hysterectomy (*n* = 5887) vs. those without (*n* = 28,024): aHR 1.27; 95% CI 1.19 to 1.35 [24]. Risk was even more pronounced in women with both hysterectomy and oophorectomy (1.9 times greater than the group without hysterectomy). In women with hysterectomy, the relative risk of hyperlipidemia was higher for women who were aged 45–64 years at the time of evaluation than for women aged ≥65 years. 

**Figure 1 jcm-10-05335-f001:**
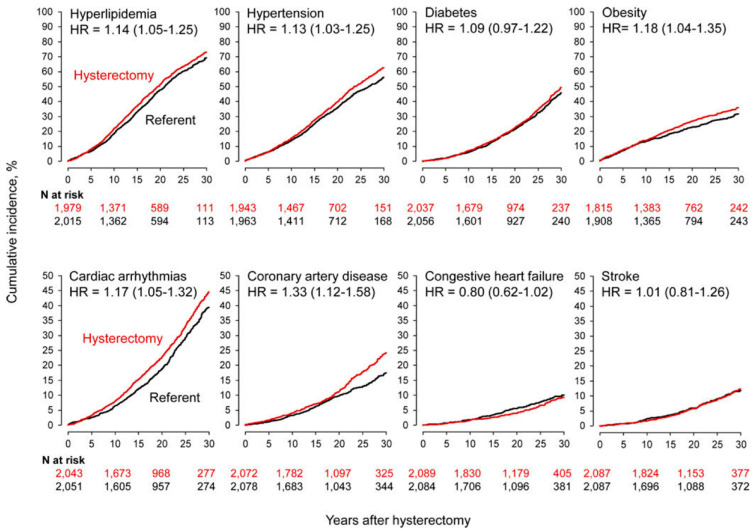
Cumulative incidence curves for cardiovascular and metabolic conditions in women who underwent hysterectomy with ovarian conservation at 35 years or younger compared with age-matched referent women (stratified analyses). The hazard ratios (HRs) and corresponding 95% confidence intervals were calculated using Cox proportional hazards models with age as the time scale and adjusted using inverse probability weights. Reprinted from *Menopause*, Cardiovascular and metabolic morbidity after hysterectomy with ovarian conservation: a cohort study, 25: 483–492, ©2018, with permission from Wolters Kluwer Health, Inc. [34].

Three studies assessed the association between hysterectomy and the risk of stroke, the results of which suggest that age at hysterectomy may be a key factor in determining risk in this outcome. In a longitudinal, national cohort study conducted using data from the Korean National Health Insurance Service (2002–2013), no association between hysterectomy and increased risk of either hemorrhagic stroke or ischemic stroke was observed in the 11,280 women with hysterectomy vs. 45,120 matched controls overall or when analyzed by age at hysterectomy [36]. Similarly, results from a nationwide population-based study conducted using the Taiwan National Health Insurance database showed no significant difference in risk of stroke in women who underwent hysterectomy (*n* = 7605) or those who did not (*n* = 30,420) [37]. However, the risk was significantly increased in women who underwent hysterectomy prior to the age of 45 years; HR 2.29 (95% CI 1.52 to 3.44). These findings were supported by results from a study of the Swedish Inpatient Registry, which showed that hysterectomy in women aged 18–49 years increased the risk of stroke, particularly with oophorectomy, compared with no hysterectomy: aHR 1.47; 95% CI 1.16 to 1.87 [35].

In summary, seven of the nine studies identified showed an increased risk of CVD, metabolic conditions, or stroke among women who had undergone hysterectomy especially when surgery was performed in women of younger age. Although no overall association between hysterectomy and mortality was observed, evidence suggests that certain subgroups may be at increased risk; future analyses will help provide more information. The link between hysterectomy and CVD is not known to be causative in these studies and the increased risk could be mediated, at least in part, through impaired ovarian function secondary to the surgery or confounding factors such as surveillance bias. Benign indications for hysterectomy may be independent risk factors for metabolic disorders such as CVD and hyperlipidemia. For example, women undergoing hysterectomy for the treatment of anovulatory bleeding from polycystic ovary syndrome, with risk factors such as obesity, metabolic syndrome, or type 2 diabetes mellitus, are at increased risk of CVD [60].

### 3.2. Impact of Hysterectomy on Cancer Risk

Four studies were identified that analyzed the association between hysterectomy and subsequent risk of developing cancer, specifically renal cell carcinoma (RCC), bladder cancer and urinary tract cancer (one study), and thyroid cancer (three studies). Two additional studies examined the incidence of cancer-related death in women with or without hysterectomy.

Altman et al. investigated the long-term risk of RCC after hysterectomy for benign indications using data from the Swedish Inpatient Registry, collected over a 30-year period (1973–2003), in a nationwide longitudinal study [41]. A significant association between RCC and hysterectomy was shown with crude RCC incidence rates of 17.4 cases per 100,000 person-years among women with hysterectomy (*n* = 184,945) and 13.1 cases per 100,000 person-years among women without hysterectomy (*n* = 657,288); adjusted overall HR of 1.50 (95% CI 1.33 to 1.69). This study also showed a small increased risk of bladder cancer and urinary tract cancer among women who had undergone hysterectomy vs. those who had not (Table 3). For all three types of cancer, the highest risk was observed up to 10 years after surgery in women who had hysterectomy aged ≤44 years (RCC: HR 2.36; 95% CI 1.49 to 3.75). However, the level of contribution of obesity to the incidence of RCC, which was not analyzed in the study, was queried by Dimmitt SB in a letter to the editor because obesity is a risk factor for both UF and RCC [61]. Altman et al. acknowledged that it is plausible that obesity may have had an impact and that further clinical and experimental research is needed [41].

Three large prospective observational studies have reported an increased risk of thyroid cancer among women who have undergone hysterectomy. A study using patient data collected from the Swedish Inpatient Registry (1973–2009) analyzed the associations between hysterectomy and occurrence of thyroid cancer subtypes (papillary, follicular, and “others”) [44]. The final study population in this analysis included 90,235 women with hysterectomy and 5,379,843 women without hysterectomy. There were nearly 3000 cases of thyroid cancer, and there was a significantly increased risk of papillary thyroid cancer observed in women who had a hysterectomy (*n* = 52) vs. those who had not had a hysterectomy (*n* = 2882) (HR 1.70; 95% CI 1.04 to 2.79). No significant association was found for follicular carcinoma or “other” thyroid cancers. In patients diagnosed with thyroid cancer, the age at initial diagnosis was lower among women with than women without hysterectomy, but survival rates did not differ [44]. Findings from a large, prospective, cohort study in France showed a twofold increased risk of thyroid cancer among women with hysterectomy (*n* = 80,197) compared with those without (*n* = 9143) (aHR 2.05; 95% CI 1.65 to 2.55) [42]. This study also showed that a history of fibroids was associated with an increased risk of thyroid cancer (aHR 1.91; 95% CI 1.50 to 2.44), even when adjusted for hysterectomy, leading authors to conclude that further examination is required to determine any shared mechanisms between UF and thyroid cancer. Finally, in a third large, prospective cohort study conducted at 40 clinical centers in the US, an increased risk of thyroid cancer was observed in postmenopausal women with hysterectomy (*n* = 24,575) vs. without (*n* = 43,139); HR 1.46; 95% CI 1.16 to 1.85 [43]. However, authors acknowledged the potential reflection of confounding by indication for hysterectomy (e.g., abnormal uterine bleeding; the association between UF and thyroid modules; or exposure to ionizing radiation during childhood) in the suggested association between thyroid cancer and hysterectomy, although they believed that detection bias was unlikely.

These findings suggest that there may be an association between hysterectomy and an increased risk of certain types of cancer, notably thyroid cancer, RCC, bladder cancer, and urinary tract cancer [41,42,43,44]. It is important to acknowledge the potential biases in these studies in that the underlying condition leading to hysterectomy (e.g., obesity, abnormal uterine bleeding, or exposure to ionizing radiation) may be associated with these cancers; therefore, the women included in the studies may be predisposed to them. More information is required to understand the association between the risk of these specific types of cancer and hysterectomy. 

### 3.3. Impact of Hysterectomy on Pelvic Anatomy and Lower Urinary Tract Symptoms

The literature search identified four studies that investigated the impact of hysterectomy on pelvic anatomy or lower urinary tract symptoms, such as dysuria, urinary retention, urinary incontinence, increased urinary frequency and urgency, in women with benign indications who underwent this procedure [45,46,47,48]. 

Li et al. retrospectively investigated the impact of hysterectomy on the risk of de novo lower urinary tract symptoms in a nationwide, population-based study of Taiwanese women conducted between 2000 and 2012 [48]. Multivariate analysis showed an increased risk of de novo lower urinary tract symptoms in women with hysterectomy (*n* = 8514) vs. age-matched controls (*n* = 34,056) without a prior hysterectomy (aHR 1.57; 95% CI 1.46 to 1.70; *p* < 0.001). When symptoms were analyzed individually, hysterectomy was associated with significantly increased risks of all symptoms, namely dysuria (aHR 1.50; 95% CI 1.22 to 1.85; *p* < 0.001), urinary retention (aHR 1.66; 95% CI 1.26 to 2.18; *p* < 0.001), urinary incontinence (UI; aHR 2.03; 95% CI 1.74 to 2.37; *p* < 0.001), and increased urinary frequency and urgency (aHR 1.41; 95% CI 1.28 to 1.56; *p* < 0.001). 

Three of the identified studies were nationwide, longitudinal, prospective analyses of data from the Swedish Inpatient Registry, collected over a 30-year period (1973–2003). The studies demonstrated an increased risk of pelvic organ prolapse surgery [46], pelvic organ fistula surgery [45], and stress UI surgery [47] with hysterectomy. In the first study, of the women with hysterectomy (*n* = 162,488), 3.2% had pelvic organ prolapse surgery compared with 2.0% of women without hysterectomy (*n* = 470,519) (HR 1.7; 95% CI 1.6 to 1.7) [46]. Furthermore, women with hysterectomy (*n* = 182,641) were four times more likely to have pelvic organ fistula surgery than those without (*n* = 525,826; HR 3.8; 95% CI 3.3 to 4.3) [45]. Finally, irrespective of surgical technique, an increased risk of stress UI surgery was observed in women with hysterectomy (*n* = 165,260) vs. age- and county-of-residence-matched controls without hysterectomy (*n* = 479,506): HR 2.4, 95% CI 2.3 to 2.5 (Figure 2) [47]. The highest overall risk was observed in the first 5 years post-hysterectomy, but excess risk remained even after 10 years [47]. 

In summary, all four studies showed that hysterectomy was associated with a negative impact on pelvic anatomy and lower urinary tract symptoms. In the studies in which an increased risk was identified, the authors highlight that this risk should be considered in the surgical decision-making process and recommend counselling patients on the risks prior to surgery [45,46,47]. It should be noted that clinical trials and large-scale prospective studies are needed to evaluate causality. It is important for the surgeon to recognize pre-existing prolapse or UI that may benefit from concomitant or joint procedures, e.g., hysterectomy with prolapse repair [62,63]. Risk of pelvic organ prolapse has been reported to occur more frequently after vaginal hysterectomy for prolapse compared with hysterectomy for non-prolapse conditions [64] and prolapse, as an indication for hysterectomy, increases risk of prolapse recurrence [65]. These studies were not included in the literature review because they do not include women with UF, but they further highlight the need for patient counselling prior to hysterectomy [65]. Finally, the risk of stress UI may increase with hormone deprivation and postmenopausal status [66,67]. The impact of hysterectomy on hormonal status is discussed in the next section.

### 3.4. Impact of Hysterectomy on Premature Menopause, Premature Ovarian Failure, Frailty, Osteoporosis, and Vasomotor Symptoms

Six studies were identified that examined the risk between hysterectomy and premature menopause, premature ovarian failure, frailty, osteoporosis, or vasomotor symptoms.

Two prospective cohort studies examined the impact of hysterectomy for benign indications on ovarian function and age of menopause through assessment of follicle-stimulating hormone levels. In these studies, an increased risk of earlier ovarian failure [49] and onset of menopause [50] was observed among women who underwent hysterectomy. In the Prospective Research on Ovarian Function Study, premenopausal women with hysterectomy without bilateral oophorectomy (*n* = 406) had an almost twofold increased risk for ovarian failure vs. premenopausal women without hysterectomy (*n* = 465) (HR 1.92; 95% CI 1.29 to 2.86), with 14.8% vs. 8.0% of women experiencing ovarian failure after 4 years, respectively [49]. Of the women in the hysterectomy group, the majority (74.4%) reported a history of UF. Risk was also significantly increased in women who retained both ovaries (HR 1.74; 95% CI 1.14 to 2.65). The authors of the study concluded that although it is unknown whether hysterectomy itself or the underlying condition indicating hysterectomy caused earlier ovarian failure, it is important to discuss these risks with patients prior to surgery. In the second study, 257 women with hysterectomy were identified from the gynecological surgical bookings system at National Women’s Hospital, New Zealand and from private gynecological practices [50]. Of these 257 women, 53 (20.6%) reached menopause during the 5-year follow-up period vs. 19 (7.3%) of 259 women without hysterectomy. Among women who underwent hysterectomy, the rates of menopause were further increased among women who had unilateral oophorectomy (*n* = 28), with 35.7% reaching menopause within 5 years of follow-up.

The impact of hysterectomy on frailty was assessed in a cross-sectional analysis of the Canadian Longitudinal Study on Aging Baseline Comprehensive Cohort (2012–2015) [51]. The frailty index was 21% higher (*p* < 0.001) in women with menopause and hysterectomy (*n* = 2182) compared with women in the normal menopause group (*n* = 4747). Premenopausal hysterectomy was also shown to increase the risk of osteoporosis in a national sample cohort study from South Korea regardless of bilateral oophorectomy status [53]. The study analyzed 9082 women with hysterectomy and 36,328 matched controls without hysterectomy. The risk of osteoporosis increased with hysterectomy (aHR 1.45; 95% CI 1.37 to 1.53; *p* < 0.001) and was highest in the subgroup of women who had hysterectomy in the youngest age category (40–44 years; aHR 1.84; 95% CI 1.61 to 2.10). Among women who underwent hysterectomy, increased risk of osteoporosis was greater for both those with bilateral oophorectomy (*n* = 1124; aHR 1.57; 95% CI 1.37 to 1.79; *p* < 0.001) and those without (*n* = 7958; aHR 1.43; 95% CI 1.34 to 1.51; *p* < 0.001) compared with the non-hysterectomized group with bilateral oophorectomy (*n* = 4496) and without (*n* = 31,832), respectively. The authors hypothesized that this may have been due to the resultant long-term decrease in estrogen secretion in women following hysterectomy, leading to greater gradual bone mineral loss and greater risk of osteoporosis, compared with women without hysterectomy. 

An association between hysterectomy and increased risk of persistent vasomotor symptoms, hot flush, and night sweats was demonstrated in an analysis of a cohort of women from the Australian Longitudinal Study on Women’s Health [54]. A higher proportion of women with hysterectomy and ovarian conservation (*n* = 1129) vs. women without hysterectomy (*n* = 4977) experienced a constant pattern of hot flushes (30% vs. 15%) and were more likely to experience constant vs. minimal hot flushes (odds ratio 1.97; 95% CI 1.64 to 2.35). Similarly, the proportion of women with hysterectomy and ovarian conservation who experienced a constant pattern of night sweats was also higher, compared with women without hysterectomy (19% vs. 9%). The former group were also more likely to experience constant vs. minimal night sweats (odds ratio 2.09; 95% CI 1.70 to 2.55). In addition, a small prospective study in New Zealand investigated gynecological and urinary symptoms, and sexual functioning among premenopausal women who had undergone hysterectomy with conservation of at least one ovary (*n* = 257) vs. those without prior hysterectomy (*n* = 257) [52]. Five years after surgery, reports of hot flushes were higher among women with vs. without hysterectomy (41% vs. 19%; *p* = 0.05), as were reports of vaginal dryness (48% vs. 20%; *p* < 0.004). There were no notable differences between the groups in the frequency of other gynecological or urinary symptoms, or in the sexual functioning parameters assessed.

In summary, all six studies showed an association between hysterectomy (both with and without oophorectomy) and an increased risk of either premature menopause, premature ovarian failure, frailty, osteoporosis, or other vasomotor symptoms. It is important that patients are aware of these risks when considering options for the management of UF.

### 3.5. Impact of Hysterectomy on Depression, Bipolar Disorder, and Dementia

In total, five studies were identified in which the impact of hysterectomy on depression (three studies), bipolar disorder (one study), and dementia (one study) was assessed.

In all three studies examining the impact of hysterectomy on depression, the risks of depressive symptoms were greater among women with prior hysterectomy vs. those without hysterectomy. In a study analyzing data extracted from a Korean Health Insurance database, the incidence of depression was reported to be higher in women with hysterectomy with or without bilateral salpingo-oophorectomy (*n* = 9971) vs. matched controls without hysterectomy (*n* = 39,884) after 73 months’ follow-up: 6.59 vs. 5.70 per 1000 person-years, respectively (aHR 1.15; 95% CI 1.03 to 1.29; *p* < 0.05) [55]. Similarly, analysis of data for 2094 women with prior hysterectomy (obtained from the Rochester Epidemiology Project records-linkage system in the US) also showed an increased long-term risk of de novo depression (aHR 1.26; 95% CI 1.12 to 1.41) and anxiety (aHR 1.22; 95% CI 1.08 to 1.38) [57]. Furthermore, the risk of depression was greater for women who underwent hysterectomy at a younger age (18–35 years) and was observed even among women with ovarian conservation [57]. The higher risk of depressive symptoms in women with (*n* = 1334) vs. without hysterectomy (*n* = 4002) was also observed in an Australian cohort study [56]. Among women with hysterectomy, the relative risk was 1.20; 95% CI 1.06 to 1.36 in women with ovarian conservation (*n* = 884) and 1.44; 95% CI 1.22 to 1.68 in women without ovarian conservation (*n* = 450). The authors concluded that the higher level of risk observed in the women with hysterectomy could not be explained by lifestyle or socioeconomic factors.

In a retrospective cohort study using patient data captured by Taiwan’s National Health Insurance Research Database, the risk of bipolar disorder was shown to be increased in women who underwent hysterectomy with oophorectomy (*n* = 4337) vs. those who did not undergo hysterectomy or oophorectomy (*n* = 17,348): adjusted incidence rate ratio 2.19; 95% CI 1.94 to 2.49 [59].

The impact of hysterectomy on dementia was evaluated in a Danish nationwide historical cohort study [58]. Although hysterectomy did not increase the overall risk of dementia, stratification of women by age at dementia diagnosis showed an increased relative risk of early onset dementia (before age 50 years) among women who underwent hysterectomy (relative risk 1.38; 95% CI 1.07 to 1.78). This was most pronounced in women who underwent hysterectomy and bilateral oophorectomy vs. those who did not undergo hysterectomy: relative risk 2.33; 95% CI 1.44 to 3.77. The authors of the study linked this to premature estrogen deficiency, to which the younger brain may be more vulnerable. These findings were supported by a recent meta-analysis of 11 studies that examined the association between surgical menopause (defined as bilateral oophorectomy prior to menopause) and the risk of dementia or decline in cognitive function [68]. The meta-analysis identified four studies in which no association was identified between surgical menopause at any age and dementia; however, early surgical menopause (≤45 years of age) was associated with a statistically significantly higher risk (HR 1.70; 95% CI 1.07 to 2.69). In contrast, the effect of surgical menopause on decline in verbal memory, semantic memory, and processing speed was not age-dependent. An additional study (which did not include a control group and is therefore not included in Table 3) assessed the risk of dementia after hysterectomy using 14-year data from the National Health Insurance Research Database in Taiwan, which highlighted an increased risk of dementia with general anesthesia versus spinal anesthesia [69], highlighting the complexity of the possible factors that impact risks associated with hysterectomy. 

In all five studies identified, an increased risk of negative impacts on mental and cognitive health of patients was observed. This risk may be lower with ovarian preservation [56,58,59]. It is important to interpret these results with some caution due to the observational nature of the studies. Additional, robust studies are required to fully evaluate the impact of hysterectomy on mental and cognitive health.

## 4. Considerations for Future Management of Our Patients with UF

The goal of UF treatment is to alleviate UF-associated symptoms, which can have a major impact on women’s lives. Hysterectomy is definitive in its results and is currently the predominant surgical treatment option for patients with UF. For some patients, it may be the only suitable option. The short-term risks of hysterectomy are well-described, as are the means of mitigating the risks, for example, by choosing the least invasive surgical route [22]. However, hysterectomy is also associated with complications that can be significant and long-lasting. Furthermore, evidence is emerging that suggests some potential long-term risks of hysterectomy. The results of this systematic literature review highlight potential risks of cardiovascular disease; hypertension and stroke; urinary tract cancer; thyroid cancer; incontinence; pelvic prolapse; pelvic organ fistula; lower urinary tract infection; ovarian failure and premature menopause, along with other consequences of estrogenic decline, including bone mineral density loss; vasomotor symptoms; frailty; depression; and a decline in cognitive function. 

It is important to acknowledge the limitations of the available data and the outcomes of this literature review in terms of how informative this is about the long-term consequences of hysterectomy, because of the inbuilt bias of observational, epidemiological studies utilized in this type of research. It is challenging to collect long-term outcome data in the setting of randomized clinical trials, especially in the context of specific questions such as the outcomes of surgical intervention versus non-surgical treatment, where ethical aspects also play an important role. The available studies examine possible associations and hypotheses rather than causality. To truly evaluate the long-term consequences of hysterectomy, the use of adequate studies, which are appropriately designed to capture relevant information, for example, using registry-based data in which women are monitored following hysterectomy, is required. However, due to the frequent use of hysterectomy compared with other alternatives, it is important to examine the available data. The search criteria used in this review were developed with the aim of identifying the highest quality data available. Most of the studies included not only patients with UF but also those with other benign conditions to maximize the available information in the area with limited data. To strengthen the robustness of the data, only studies with comparative cohorts were included.

Although more adequate evidence is required before any definite conclusions can be reached, it may be useful to consider the current findings during the treatment decision-making process and incorporate some discussion into the counselling process. When exploring the optimal UF management option, we should enter into a shared decision-making process with our patient [70]. To enable this, our patient must be aware that there is a joint decision to be made and, as with any consenting surgical procedure, they must be fully informed of the benefits and risks of all options available to them. The potential benefits should outweigh the potential risks of the surgery. Otherwise, surgery, especially hysterectomy, should be deferred, and other options must be strongly considered. It is important for us to consider the following: the individual requirements (including disease and fertility status), wishes, and opinions of our patients; our personal clinical experience and the experience of our peers. Open communication and counselling of patients are vital to the process of reaching a shared understanding of the problem and treatment options. Similarly, continued education on HCPs will help to ensure that informed treatment strategies are developed. The currently available guidelines focus on surgery, and there is a need for updated recommendations on the UF treatment algorithm that consider both recent advances in medical treatments and the risk–benefit profile of all options.

With the development of new therapeutic options, including long-term, well-tolerated medical treatments [71,72,73,74,75], the paradigm will potentially change, and give patients the opportunity to use long-term medical therapies indicated specifically for UF. Assessing and communicating the benefits and risks of all choices for women with UF will remain key to ensuring that our patients receive the best available treatment to meet their individual needs.

## Figures and Tables

**Figure 2 jcm-10-05335-f002:**
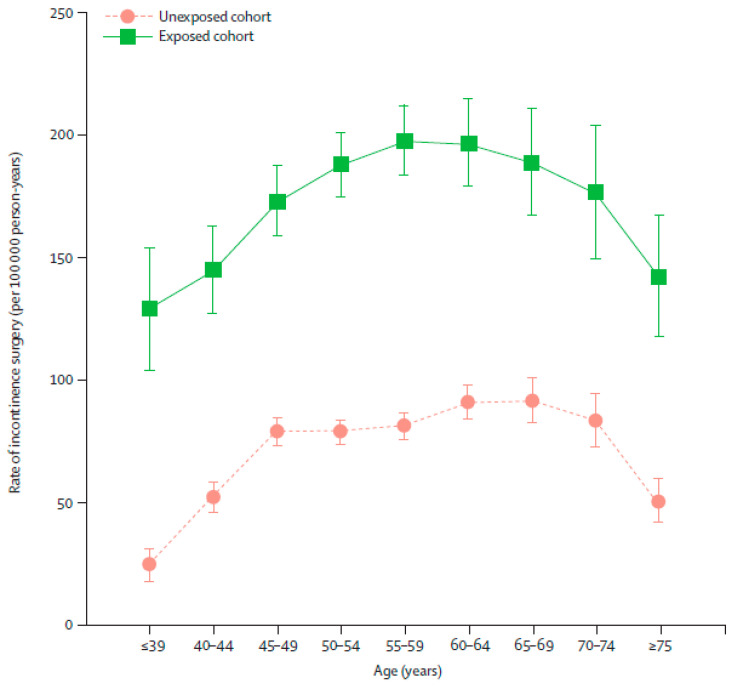
Age-specific rates of first urinary incontinence operation in women in the exposed and unexposed cohorts. Age-specific rates are shown with 95% CIs. Age intervals show attained age during follow-up period. Reprinted from *The Lancet*, 370, Altman et al., Hysterectomy and risk of stress-urinary-incontinence surgery: nationwide cohort study, 1494–1499, ©2007, with permission from Elsevier [47].

**Table 1 jcm-10-05335-t001:** Effect of hysterectomy approach on the risk of major morbidities and mortality. Reprinted by permission from Amir Wiser et al.: Springer Nature, Gynecological Surgery Abdominal versus laparoscopic hysterectomies for benign diseases: evaluation of morbidity and mortality among 465,798 cases, Amir Wiser et al., 2013 [20].

Outcome	Abdominal (*n* = 389,189), n (%)	Laparoscopic (*n* = 76,609), n (%)	OR (95% CI)	*p* Value
DVT	2879 (0.74)	502 (0.66)	0.88 (0.80–0.96)	0.04
PE	3099 (0.8)	522 (0.68)	0.85 (0.77–0.93)	0.006
DVT or PE	3281 (0.84)	529 (0.69)	0.48 (0.24–0.95)	0.0004
Blood transfusion	18,124 (4.7)	1805 (2.4)	0.56 (0.42–0.74)	0.0001
Bowel perforation	490 (0.13)	52 (0.07)	NA	0.0001
Bladder injury	17 (<0.01)	0 (0.0)	0.29 (0.27–0.31)	NA
Acute MI	133 (0.03)	13 (0.02)	0.58 (0.55–0.61)	0.2
Length of stay >6 days	15,917 (4.1)	804 (1.0)	0.29 (0.27–0.31)	0.0001
Death	123 (0.03)	9 (0.01)	0.69 (0.39–1.2)	0.036

CI, confidence interval; DVT, deep vein thrombosis; MI, myocardial infarction; NA, not available; OR, odds ratio; PE, pulmonary embolism.

**Table 2 jcm-10-05335-t002:** Inclusion criteria used for selection of studies evaluating long-term consequences of hysterectomy.

Study Inclusion Criteria	
Patients	● Patients with UF ● Patients treated with hysterectomy for UF
Interventions	● Hysterectomy (with or without unilateral/bilateral oophorectomy)
Comparator	● No hysterectomy, external control group, healthy controls
Outcomes	● Long-term (≥3 years after hysterectomy) consequences, including morbidity and mortality outcomes, adverse events and safety issues, associated conditions, or need for further surgery
Study design	● Observational studies (including cohort studies, real-world evidence studies, registry studies), follow-up studies, open-label extension studies, non-randomized clinical studies, prospective studies, or retrospective studies including ≥100 patients
Countries	● Not restricted by country or region
Date restriction	● Studies published between January 2005 and December 2020
Publication type	● Full-text journal articles only
Language restriction	● Not restricted by publication language

UF, uterine fibroids.

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
