# Peer review of "What We Know about the Long-Term Risks of Hysterectomy for Benign Indication—A Systematic Review"

_jcm, 2021, doi:10.3390/jcm10225335_

Round 1
Reviewer 1 Report
Dear authors,
I have reviewed your manuscript entitled “What do we know about the long-term risks of hysterectomy in uterine fibroids?“.
Overall this review is an interesting one.
The topic is relevant, but is not new. However, the manuscript should be further improved.
Major revision will need to be performed, however, to address the following comments. In general, the authors should clarify the following points:
- It may be of great interest the conception of an flowcharts to help readers.
- What is the novelty of this study? It is mandatory to explain this in the manuscript.
- If possible authors should try to obtain a better quality figures 1.
- Please add the title and explanation of the figures below (see instruct for auth).
- Knowing these results, what are the conclusions with regard to improving care?
- Generally, the manuscript needs some minor punctuations editing.
Reviewer 2 Report
Thanks for letting me the opportunity to review this article entitled « what do we know about the long term risk of hysterectomy in uterine fibroids?
This article is well written and gives to the reader some major elements about this problematic.
A lot of work has been performed to produce this detailed review.
However there are some points that need to be clarified in order to enhance this manuscript.
First of all in the title, why not giving the information that a systematic review has performed. This is major information that needs to be presented to the reader.
Introduction line 45, infertility should also be cited as you speak for symptoms caused by UF.
Line 49-50: please provide some examples about non-surgical and minimally invasive procedures, I think you want to talk about embolization or some hysteroscopic procedures?
Line 53 54, please provide more information about the database and the estimation that you provide here (which year, estimation from national databases?).
Also what the rate of minimally invasive procedures of UF
Line 60-61: please give more details about “substantial improvements”: less symptoms, evaluated how?
In this introduction you don’t speak about the occurrence of oophorectomy performed or not during the surgical. You should make a statement about this and that long term consequences of hysterectomy may depend of the occurrence of bilateral oophorectomy or not.
Methods:
You choose to begin after 2005 and stated that this is to account for potential differences in hysterectomy techniques with surgical advances in the past surgical decades” What kind of differences are relating for? Use of laparoscopy? Use of robotic surgery? Use of new coagulation energy. Please provide some explanations here.
Why you chose to include only studies with at least 100 subjects?
It should be wise when you perform a systematic review, to follow to PRISMA guidelines statement (that has been updated recently). As I read your methods, I think you have fulfilled them. Please state if you have performed your review following PRISMA, and also state if your review have been registered into the PROSPERO database.
Results:
Please state at the beginning of the results that results and discussion of each aspect of the review have been matched together (often, results are presented without any interpretation which is not the case here).
Line 123., what are the potential confounders that have been taken into account.
Line 187-189: this belongs to the conclusion
Line 218-219: I don’t understand your results. There is a group comparing 52 hysterectomies vs. 2882 non hysterectomy? Please provide more details, how many HT vs how many non HT are compared and the rate of thyroid cancer.
Line 242: please provide what are the lower urinary tracts symptoms you are referring to.
Considerations:
Line 401-402: what kind of higher quality studies? (For examples registries with women followed after HT could be an option?).
Round 2
Reviewer 1 Report
Dear author’s
I think that in this format your article is ready for publication.
This manuscript is a resubmission of an earlier submission. The following is a list of the peer review reports and author responses from that submission.
Round 1
Reviewer 1 Report
The title does not reflect the manuscript and needs to be changed.
Authors 1 and 3 come from the same department therefore they do not require different affiliations.
This manuscript provides the reader with a lot of statistics regarding different options, complications and outcomes for women with fibroids in the US. A systematic review regarding long term outcomes in women with or without hysterectomy for fibroids is then presented.
The title does not reflect the very lengthy manuscript and potentially the manuscript could be presented as two separate manuscripts rather than in its current format. Data on the US and a systematic review, both together are not necessary.
The paper although contains a lot of information does not flow/link together providing the reader with a lot of potentially unnecessary information irrelevant to their place of work.
Abstract –
The abstract does not fully describe the manuscript but seems to focus on the systematic literature review, ignoring all the data made available in the first four pages.
Introduction
- Need to define further what hysterectomy vs minimally invasive includes. Different types of hysterectomy and different minimal invasive treatments include radiological and outpt fibroid treatment/ inpt fibroid treatment with myosure for example vs open myomectomy.
- Reference needed for line 57, discussing menstruation.
- Line 63, diagnosis cannot truly be made through examination alone, you may consider the woman to have fibroids but a palpated mass could be numerous different pathologies, I think imaging would be required for a diagnosis.
Why is hysterectomy performed so frequently in patients with UF in the US?
- Although the title for this section is as above the section doesn’t really answer the question posed. The section documents what is done in the US but not why as such.
- This data only documents US facts and figures meaning it can not be generalised and alienates a large number of readers.
- Line 136 – Is this just in the US as the use of myosure in an outpt setting would be seen very commonly in the UK. It is difficult to differentiate when you discuss minimally invasive exactly what you are talking about, radiological treatment/ ambulatory treatment.
- Line 167 – this is very vague, be more specific, what medical treatments? Coil? Pills? GnRH?
What do we know about the complications associated with hysterectomy, especially the associated LT risk?
- 231 – minimally invasive hysterectomy is discussed here. The use of minimally invasive versus non-minimally invasive for hysterectomy and other treatments needs to be made obvious to the reader, the word is used interchangeably to describe different procedures. The manuscript needs to be specific from the start.
- 234 – Again all US data, the reader needs to be made aware of this, not generalisable data.
Reviewer 2 Report
Article review (Management of uterine fibroids: to operate or not to operate? jcm-1151076):
Summary: This is a systematic review of the medical and surgical management of uterine fibroids. The authors provide an overview of medical and surgical treatment options for symptomatic fibroids and suggest that hysterectomy is associated with significant long-term risks that should be considered when treatment decisions are made. Overall, the evidence that is being presented is subject to significant bias and therefore misleading.
Comments/Revisions
- The authors cite several population-based studies that suggest an association of hysterectomy with increased risk for many medical problems, such as cardiovascular disease, hypertension, stroke, depression, and dementia. This is not high-quality evidence for several reasons. First, these population-based retrospective studies are subject to significant bias. For instance, patients with any type of intervention, including a hysterectomy, are more likely to be health-conscious and have regular screening visits that would lead to a more frequent diagnosis of various medical problems. Second, the accuracy of documentation in these studies regarding the removal of the ovaries is questionable. Furthermore, the studies that looked at all-cause mortality found no association with prior hysterectomy. Lastly, there is a complete lack of evidence on the effect of the alternative treatment options for fibroids on these medical problems.
- The authors cite retrospective studies that suggest an association between hysterectomy and increased risk for urinary tract and thyroid cancer. These studies are also subject to significant bias. It would be more reasonable to look at the incidence of cancer of any type and overall cancer-related mortality that would also consider gynecologic malignancies such as uterine, ovarian, and cervical cancer.
Recommendation
Reject